# The Importance of Epigallocatechin as a Scaffold for Drug Development against Flaviviruses

**DOI:** 10.3390/pharmaceutics15030803

**Published:** 2023-03-01

**Authors:** Mônika A. Coronado, Ian Gering, Marc Sevenich, Danilo S. Olivier, Mohammadamin Mastalipour, Marcos S. Amaral, Dieter Willbold, Raphael J. Eberle

**Affiliations:** 1Institute of Biological Information Processing (IBI-7: Structural Biochemistry), Forschungszentrum Jülich, 52428 Jülich, Germany; 2Priavoid GmbH, Merowingerplatz 1A, 40225 Düsseldorf, Germany; 3Integrated Sciences Center, Campus Cimba, Federal University of Tocantins, Araguaína 77824-838, TO, Brazil; 4Institut für Physikalische Biologie, Heinrich-Heine-Universität Düsseldorf, Universitätsstraße, 40225 Düsseldorf, Germany; 5Institute of Physics, Federal University of Mato Grosso do Sul, Campo Grande 79070-900, MS, Brazil; 6JuStruct: Jülich Centre for Structural Biology, Forschungszentrum Jülich, 52428 Jülich, Germany

**Keywords:** epigallocatechin gallate, EGCG, epigallocatechin, EGC, epigallocatechin gallate octaacetate, NS2B/NS3 protease, DENV2, YFV, WNV, ZIKV

## Abstract

Arboviruses such as Dengue, yellow fever, West Nile, and Zika are flaviviruses vector-borne RNA viruses transmitted biologically among vertebrate hosts by blood-taking vectors. Many flaviviruses are associated with neurological, viscerotropic, and hemorrhagic diseases, posing significant health and socioeconomic concerns as they adapt to new environments. Licensed drugs against them are currently unavailable, so searching for effective antiviral molecules is still necessary. Epigallocatechin molecules, a green tea polyphenol, have shown great virucidal potential against flaviviruses, including DENV, WNV, and ZIKV. The interaction of EGCG with the viral envelope protein and viral protease, mainly identified by computational studies, describes the interaction of these molecules with viral proteins; however, how the viral NS2B/NS3 protease interacts with epigallocatechin molecules is not yet fully deciphered. Consequently, we tested the antiviral potential of two epigallocatechin molecules (EGC and EGCG) and their derivative (AcEGCG) against DENV, YFV, WNV, and ZIKV NS2B/NS3 protease. Thus, we assayed the effect of the molecules and found that a mixture of the molecules EGC (competitive) and EGCG (noncompetitive) inhibited the virus protease of YFV, WNV, and ZIKV more effectively with IC_50_ values of 1.17 ± 0.2 µM, 0.58 ± 0.07 µM, and 0.57 ± 0.05 µM, respectively. As these molecules fundamentally differ in their inhibitory mode and chemical structure, our finding may open a new line for developing more effective allosteric/active site inhibitors to combat flaviviruses infection.

## 1. Introduction

The *Flaviviridae* family consist of 89 virus species divided into four genera. The genus *Flavivirus* encompasses 53 enveloped positive-sense single RNA viruses. The virion consists of the three structural proteins (capsid (C), membrane (M), and envelope (E)) and seven nonstructural proteins presented in infected cells (NS1, NS2A, NS2B, NS3, NS4A, NS4B, and NS5) [1,2,3].

Classified as mosquito-borne flaviviruses, Dengue virus (DENV), yellow fever virus (YFV), and Zika virus (ZIKV) are mainly transmitted by *Aedes* spp. [4] and West Nile virus (WNV) is transmitted by *Culex* spp. [5] DENV, YFV, and ZIKV are responsible for causing endemics or epidemics worldwide in tropical/subtropical regions [6], and WNV is endo/epidemic in Europe [7].

Mosquito-borne viruses are highly suited to the ‘one health’ concept for combating infectious diseases, demonstrating the interdependence of human, animal, and ecosystem health [8]. Mosquito-borne viruses are categorized as zoonotic diseases, accounting for approximately 75% of emerging infectious diseases [9], with more than 800 human pathogens [10]. Wild animals are victims of those pathogens, but in some cases, animals only act as reservoirs. Transmission can occur through direct contact with infected animals, such as through food consumption or as pets.

Intense media coverage of SARS [11], West Nile virus infection in the western hemisphere [12], and other infectious diseases have brought public attention to the health risk of zoonoses and other emerging or re-emerging diseases.

Increased human mobility, population growth, the exploitation of a previously undisturbed ecosystem and climate change constitute major risk factors leading them to contact new pathogens and their vectors, consequently altering the behavior, migration, and interaction of pathogens and their hosts. A glaring example is the significant spread and establishment of vector-borne zoonotic diseases worldwide, such as WNV, Crimean–Congo hemorrhagic fever (CCHF), Japanese encephalitis virus (JEV) [13], and the appearance of Chikungunya virus in Italy in 2007 [3].

Human infection caused by flavivirus ranges from mild illness (flu-like febrile illnesses with a headache, myalgia, arthralgia, and a rash without long-term consequences) to severe and life-threatening disease, which is classified into two categories, visceral and neurotropic [14].

Among the viral proteins, NS3 is a multifunctional protein with N-terminal serine protease activity. However, the protease is a complex of two subunits, the already described Nter domain (~184 aa) and an essential cofactor (~40 aa) within the viral NS2B protein [15,16]. Highly conserved among flaviviruses, it is an essential enzyme in virus replication. Given its essential role, the NS2B/NS3 proteases (NS2B/NS3^pro^) of flaviviruses are promising drug targets; for example, DENV2, YFV, WNV, and ZIKV NS2B/NS3^pros^ present high amino acids sequence conservation and all four have the same structural fold, which confers a similar substrate specificity profile [16,17]. Due to their importance, we selected all four viruses to study the effect of a bioactive plant derivative molecule against NS2B/NS3^pro^.

To obtain much deeper insights and to identify a broad-spectrum molecule, we performed a combinatorial approach to identify how a small, potent molecule derived from green tea is able to inhibit the selected NS2B/NS3^pro^ of DENV2, YFV, WNV, and ZIKV, which may serve as a lead compound to develop a potent antiviral drug.

Epigallocatechin gallate (EGCG) and epigallocatechin (EGC) are polyphenols found in green tea; several studies have shown a broad-spectrum of antiviral activity against viral entry into host cells [18,19]. Li and collaborators [16] showed the inhibition of ZIKV NS2B/NS3^pro^ by EGCG and ECG (epicatechin gallate) with an IC_50_ value of 87 and 89 µM, respectively. It was also shown that EGCG inhibits ZIKV entry in Vero E6 cells [20]. Recently, Liu and coll. demonstrated the effect of EGCG against SARS-CoV-2 inhibiting the spike protein from binding to the ACE2 receptor [21] while reducing coronavirus replication in a mouse model [22].

In this report, we have explored the in vitro and in silico inhibitory capacity of EGC, EGCG, and EGCG octaacetate (a modified form of EGCG) against NS2B/NS3 protease of DENV2, YFV, WNV, and ZIKV.

## 2. Materials and Methods

### 2.1. Molecules

All the molecules used in this study were purchased from EGCG (Merck, Darmstadt, Germany; ≥95%), EGC (Biomol, Hamburg, Germany; 98%), and AcEGCG (Biomol, Hamburg, Germany; 98%).

### 2.2. NS2B/NS3 Protease Expression and Purification

The DENV2 and WNV NS2B/NS3^pro^ constructs were provided by Prof. Rolf Hilgenfeld, University Lübeck, Germany. The cDNA encoding DENV2 and WNV NS2B/NS3^pro^ (GenBank Protein Accession number AHZ13508.1 and AAA48498.2) was synthesized (Fisher Scientific-Geneart, Regensburg, Germany) and implemented in the ampicillin-resistant vector pET15b (+) (Fisher Scientific GeneArt, Regensburg, Germany). The codon-optimized cDNA encoding YFV and ZIKV NS2B/NS3^pro^ (GenBank Protein Accession number AAY34247.1, isolate Angola/14FA/1971, AMA12085., respectively) was synthesized and implemented in the kanamycin-resistant vector pET-24a(+) (BioCat GmbH, Heidelberg, Germany) and pET-28a(+) (Fisher Scientific GeneArt, Regensburg, Germany), respectively. The construct contains the NS2B cofactor region (DENV2: residues 1394–1440; YFV: 1402–1449; ZIKV: 1421–1467; WNV: 1420–1465) linked with NS3 protease (DENV2 residues 1476–1660; YFV: 1491–1660; ZIKV: 1503–1672; WNV: 1502–1671) by a GGGGSGGGG linker. A single chain NS2B/NS3 enzyme of DENV2, YFV, WNV, and ZIKV was overexpressed in *E. coli* cells (DENV2, WNV and YFV/(Lemo21(DE3)); ZIKV/(BL21(DE3)T1) with a construct containing an N-terminal hexahistidine affinity tag and a TEV protease cleavage site (ENLYFQG) [23,24]. Cells were harvested, lysed, and centrifuged at 7000 rpm for 90 min. The supernatant was collected and applied into an affinity column using immobilized metal (Ni-NTA). The target protein eluted from 100 mM to 500 mM imidazole. Subsequently, the virus proteases were further purified using a Superdex 75 10/30 column. The purity (95%) was assessed by SDS-PAGE [23,24].

### 2.3. Enzymatic Assay

A previously described enzymatic assay [23,24] was used to investigate the inhibitory activity of EGC, EGCG, and EGCG octaacetate (AcEGCG) molecules against DENV2, YFV, WNV, and ZIKV NS2B/NS3^pro^.

The catalytic activity was performed using a fluorogenic substrate: Boc–Gly–Arg–Arg–AMC; BACHEM, Bubendorf, Switzerland (DENV2, YFV, and WNV) and Pyr–Arg–Thr–Lys–Arg–AMC; BACHEM, Bubendorf, Switzerland (ZIKV) via a time-resolved fluorescence resonance energy transfer (TR-FRET) method in buffer containing 50 mM Tris-HCL pH 8.5; 20% glycerol; 0.01% Triton X-100 (DENV2 and YFV) and 20 mM Tris-HCL pH 8.5; 10% glycerol; 0.01% Triton X-100 (ZIKV and WNV). The virus proteases were incubated at room temperature with the concentration of the individual molecules (Table 1) in a Corning 96-Well plate (Sigma-Aldrich, St. Louis, MO, USA). After 60 min, a final concentration of 50 µM (DENV2, WNV, and YFV) and 20 µM (ZIKV) of the substrate was added, and a fluorescent signal was acquired using an Infinite 200 PRO plate reader (Tecan, Männedorf, Switzerland) at 37 °C. The excitation and emission wavelengths were 380 nm and 460 nm, respectively. Inhibition assays were performed in triplicate. In order to determine the IC_50_ value, a dose-response curve was plotted, and the IC_50_ determination was based on a regression plot.

### 2.4. Determination of Inhibition Mode

Different final concentrations of the molecules and the substrate were used to determine the mode of inhibition. DENV2, YFV, WNV, and ZIKV proteases were pre-incubated (separately) with the molecules (EGC, EGCG, and AcEGCG) at different concentrations at room temperature for 30, 60, and 90 min (Table 2). The reaction was measured by adding the corresponding concentration series of the substrate over 30 min with intervals of 60 s at 37 °C. The data analysis was performed using a Lineweaver–Burk approach, comparing the reciprocal of velocity (1/V) vs. the reciprocal of the substrate concentration [25,26]. All of the measurements were performed in triplicate, and data are presented as mean ± SD.

### 2.5. Fluorescence Spectroscopy

All of the measurements were performed in buffer containing 20 mM Tris-HCl pH 7.5; 150 mM NaCl; 4% Glycerol (DENV2) and 25 mM Tris-HCl pH 8.5; 150 mM NaCl; 5% Glycerol (YFV, ZIKV, and WNV) at RT in triplicate. Epigallocatechin molecules shift excitation change studies were performed by adding aliquots from a stock solution into the cuvette containing 50 µL of the protein with a final concentration of 5 µM (Table 3). Fluorescence intensities were corrected for any dilution effects. During the interaction studies of the molecules with NS2B/NS3^pro^, the protein solution within the cuvette was stepwise titrated with the molecule’s stock solution (0.5 mM + 5 μM protein). After each titration, measurement was conducted.

The intrinsic Trp fluorescence was measured with a QuantaMaster40 spectrofluorometer (PTI, Birmingham, AL, USA) using 1 cm path-length quartz cuvettes (105.253-QS, Hellma, Mühlheim, Germany). All spectra were corrected for background intensities by subtracting the spectra of pure solvent measured under identical conditions. Both excitation and emission bandwidths were set at 8.0 nm, and the excitation wavelength at 295 nm was chosen since it provides no excitation of tryptophan residues. The emission spectrum was collected in the 300–500 nm range with an increment of 1 nm.

### 2.6. Determination of Dissociation Constant Using Surface Plasmon Resonance

The dissociation constant (K_D_) of EGC, EGCG, and AcEGCG toward NS2B/NS3 proteases was determined using a Biacore T200 Surface Plasmon Resonance (SPR) instrument (Cytiva, Marlborough, MA, USA). The purified NS2B/NS3^pros^ were diluted to 50 µg/mL in 10 mM sodium acetate pH 5.0 (Merck, Darmstadt, Germany). The proteases were immobilized on separated channels on a series S CM5 sensor chip (Cytiva, Marlborough, MA, USA) by amine coupling. The flow cells on each channel were activated using 50 mM N-hydroxysuccinimide (NHS) and 16.1 mM N-ethyl-N’-(dimethyl aminopropyl) carbodiimide (EDC) (XanTec, Düsseldorf, Germany) for 7 min. The target protein was injected (separately) to a signal level of ~3000 RU. After the immobilization, each channel’s ligand and reference flow cells were quenched by a 7-min injection of 1 M ethanolamine pH 8.5 (XanTec, Düsseldorf, Germany). The K_D_ was assessed by injection of varying concentrations of the molecules (See Appendix A) at a flow rate of 30 min^−1^ at 25 °C. The analyzed molecules were primarily prepared as 10 mM stocks in DMSO or water. The K_D_ multicycle experiments were determined with 10 mM MES pH 6.1 and 150 mM NaCl as a running buffer. The stock molecules dissolved in DMSO were diluted in a buffer containing 10 mM MES pH 6.1, 150 mM NaCl, and 2% DMSO; the same was used as a running buffer.

All of the samples were injected over the flow cells for 240 s, followed by a dissociation phase of 1200 s with a running buffer. A regeneration step was performed to ensure complete dissociation of the analytes with 10% DMSO in water. The response curves of various analyte concentrations were fitted to the two-state model implemented in the Biacore T200 Evaluation software for data evaluation described by the following Equation (1) [27].
(1)$$A+B\;\begin{array}{c} \overset{ka1}{\to }\\ \underset{kd1}{\leftarrow } \end{array}\;AB\;\begin{array}{c} \overset{kd1}{\to }\;\\ \underset{kd2}{\leftarrow } \end{array}\;ABx$$

The equilibrium constants of each binding step are *k*1 = *ka*1/*kd*1 and *K*2 = *ka*2/*kd*2, and the overall equilibrium binding constant is calculated as *ka* = *kl* (*l* + *k*2) and *kd* = 1/*ka*. In this model, the analyte (*A*) binds to the ligand (NS2B/NS3^pro^) (*B*) to form an initial complex (*AB*) and then undergoes subsequent binding or conformational changes to form a more stable complex (*ABx*) [27]. The experiments were performed in duplicate, and the results are shown as the mean ± STD.

### 2.7. Circular Dichroism Spectroscopy

Seven repeated scans were performed for all CD measurements, and three separate scans were used to establish the baseline. The wavelength range applied for far-UV spectra was settled from 200 nm to 260 nm in a time constant of 1 s and 50 nm/min continuous scanning mode, using a Jasco J-1100 CD spectrometer (Jasco, Tokyo, Japan). NS2B/NS3^pro^ of DENV2, YFV, WNV, and ZIKV were separately diluted in 20 mM K_2_HPO_4_/KH_2_PO_4_ pH 7.5 to a concentration of 50 µM to investigate the influence of EGC, EGCG, and AcEGCG on the secondary structure of the proteins. The proteases were incubated with 25 µM (EGC and EGCG) and 5 µM (AcEGCG) on ice for 10 min prior to the measurements. The results are presented in molar ellipticity [**θ**] according to Formula (2):
[**θ**] = **θ**/(**c** × **1** × **10** × **n**)
(2)
where **θ** is the ellipticity measured at a given wavelength λ (deg), **c** is the protein concentration (mol L^−1^), **l** is the cell path length (cm), and **n** is the number of amino acids.

### 2.8. Molecular Dynamics and Computational Analysis

#### 2.8.1. Ligand Parameterization

The ligand structure of EGC, EGCG, and AcEGCG was retrieved from the Zinc database [28]. Ligands were geometrically optimized, and the electrostatic potentials were calculated for molecular dynamics (MD) simulation using Gaussian16 [29] at the B3LYP/6–31G* level of theory. Antechamber [30] was used to determine the restrained electrostatic potential (RESP) charges, and a general amber force field (GAFF) [31] was used for missing parameters.

#### 2.8.2. System Preparation

In silico studies were carried out on X-ray models of Dengue2 (PDB code: 4M9T), yellow fever (PDB code: 6URV), West Nile (PDB code: 2IJO), and Zika (PDB code: 5LC0) NS2B/NS3 protease. Furthermore, the docking process was initiated by following a series of necessary steps such as ligands and proteins preparation, grid generation, and molecular docking.

All of the proteins had the amino acid side chain corrected to the protonation state of pH 7.4, determined by the H++ web server [32]. The complexes were embedded in octahedral boxes with TIP3P water extended at least 10 Å from the solute surface and charge neutralized with Na^+^ or Cl^−^ counterions.

#### 2.8.3. Starting Structures

The MD and computational analysis were performed based on a previously described approach [23,33]. In order to obtain refined and relaxed structures, the models were subject to 200 ns of MD simulation; subsequently, clustering analyses were carried out to obtain a stable and representative model.

To gain insight into the binding mode of the ligand, we docked the ligands to the stable models prepared previously. Docking calculations were performed using AutoDock Vina 1.1.12 [34,35]. The AutoDockTools program [35] added polar hydrogens and partial charges to the protein and rotational bonds in the ligands. The search space was defined in the grid box, and as output results, several poses were ranked according to the scoring function of Autodock Vina.

#### 2.8.4. Molecular Dynamics Simulations Setup

AMBER18 [36] program package was used to run all of the MD simulations. The atomic interactions for the protein were described with FF19SB [37] force field, while GAFF and RESP charges were used to describe the ligands. Atomic bad contacts were eliminated from the starting structures by two rounds of energy minimization (EM). The first EM round was performed with the 5000 steepest descent steps followed by 5000 conjugate gradients, while the complex was constrained with a force constant of 10.0 kcal/mol-Å2. The second round was an unconstrained EM performed with 10,000 steps. Following EM, the complexes were linearly heated from 10 to 298 K during 500 ps under a canonical (NVT) ensemble, with the complex restrained with a force constant of 10 kcal/mol-Å2. After that, the equilibration step was performed using an isothermal-isobaric (NPT) ensemble for 6 ns, with a decreasing restrained force constant from 10 to 0 kcal/mol-Å2. Finally, the production runs, 100 ns, performed in an NVT ensemble without restraints. Langevin coupling was used to control the temperature (298 K). All of the complexes were replicated using the same starting atom position with different random initial velocities according to the Maxwell–Boltzmann distribution. The SHAKE constraints were applied to all bonds involving hydrogen atoms to allow a 2 fs dynamics time step. Long-range electrostatic interactions were calculated by the particle-mesh Ewald method (PME) using 8-Å cutoff [38].

#### 2.8.5. Structural and Molecular Dynamic Simulation Analysis

The CPPTRAJ [39] program of AmberTools19 [36] was used to analyze the MD simulations. Root mean square deviation (RMSD) and the protein backbone’s radius of gyration (Rg) were calculated to determine system stability, equilibration, and convergence. Protein flexibility was assessed by root mean square fluctuation (RMSF) for all C_α_ atoms, residue-by-residue over the equilibrated trajectories. Clustering analysis was performed with the k-means method ranging from 2 to 6, and Davies-Bouldin index (DBI) values and silhouette analyses were used to assess the clustering quality. The interaction energy between the proteins and the ligands was calculated using the generalized Born GB-Neck2 [40] implicit solvent model. Molecular mechanics/generalized Born surface area (MM/GBSA) energy was computed between the protein/ligand in a stable regime comprising the last 50 ns of the MD simulation, stripping all the solvents and ions.

## 3. Results

### 3.1. Characterization of DENV2, YFV, WNV, and ZIKV Protease Inhibition by EGC, EGCG, and AcEGCG

Analysis of the 3D model structure of the individual proteins relative to each other revealed low RMSD values, and the average value was calculated to be 0.84 Å. The values indicated that the protein structures are very similar. Using the heterogeneously expressed protein in *E. coli* cells (as described before), we tested the impact effect of two plant-derivative catechins (EGC and EGCG) and the modified one, AcEGCG (Figure 1), against the NS2B/NS3^pros^ of DENV2, YFV, WNV, and ZIKV.

Inhibition assays were carried out in triplicate, and the enzyme was pre-incubated for 60 min with varying concentrations of the molecules (Table 4) in buffer containing 50 mM Tris-HCL pH 8.5; 20% glycerol; 0.01% Triton X-100 (DENV2, YFV) and 20 mM Tris-HCL pH 8.5; 10% glycerol; 0.01% Triton X-100 (ZIKV, WNV). At the highest molecule concentration, the protease was inhibited by 100% (Appendix A).

The IC_50_ values and the inhibitory mode of the molecules toward the proteases were determined regarding the complete inhibition of protease activity (Appendix A, Table 5). Despite sharing a similar core scaffold, the molecules present a particular structure, which shows distinctive inhibitory effects. Among the tested molecules, EGCG showed the highest inhibitory activity against WNV protease with an IC_50_ value of 1.8 µM, followed by ZIKV protease with an IC_50_ value of 4.5 µM. The latter IC_50_ deviates from previous results published by Lim and colleagues (IC_50_: 87 µM) [41]. The same molecule showed IC_50_ values of 6.3 µM (DENV2) and 8 µM (YFV). Compared to EGCG, EGC does not have a trihydroxybenzoate ring, which may contribute to its inhibitory activity toward the active site. Therefore, the IC_50_ values increased drastically against DENV2 (145 µM), followed by YFV, WNV, and ZIKV (Table 5). In contrast, the addition of eight-ester acetate, AcEGCG (Figure 1), did not improve the inhibition effect and showed the weakest inhibitory effect against the proteases with an IC_50_ value of 102 µM against ZIKV NS2B/NS3^pro^ followed by WNV protease (Table 5). Similar inhibition activity against the protease was identified for DENV2 and YFV (Appendix A, Table 5).

The mechanism of inhibition was tested enzymatically with varying concentrations of both the inhibitor and the substrate. The analysis showed that EGCG and AcEGCG inhibit DENV2, YFV, ZIKV, and WNV NS2B/NS3^pro^ does not alter the Km but changes the Vmax, indicating that both molecules inhibit the proteases by a noncompetitive mode, which means allosterically (Appendix A, Table 5).

The molecules EGC and EGCG bound in different regions of the four studied viral NS2B/NS3^pros^. To monitor the effect of both molecules simultaneously, we performed a combined inhibitory assay, where both molecules (EGCG and EGC) were mixed with a molar ratio of 1:1 (Table 6. Appendix A). Increased inhibition capacity was observed for YFV, ZIKV, and WNV; however, it did not show effectiveness for DENV2 compared to the IC_50_ of the single molecule.

### 3.2. Binding Affinities of the Epigallocatechin Molecules

In order to determine the binding affinity of EGC, EGCG, and AcEGCG molecules, we used the SPR (surface plasmon resonance) approach. After activation of carboxylic groups via EDC/NHS, NS2B/NS3^pro^ was adsorbed onto the CM5 sensor chip through covalent amide binding. Finally, ethanolamine was used to block the surface’s unreacted sites. The individual molecule was passed over a CM5 sensor chip pre-immobilized with the individual protease (NS2B/NS3).

Kinetic fitting revealed dissociation constants from the low µM to low nM range for the two-state kinetic reaction, as seen in Appendix A (CD spectroscopy). The equilibrium rate constant (Kd) values in µM are shown in Table 7.

### 3.3. Fluorescence Spectroscopy

NS3B/NS3^pro^ interaction with the epigallocatechin molecules induced a time-dependent secondary state, as demonstrated by SPR (Appendix A) and CD (Appendix A) experiments; for that reason, we checked the potential of underlying second-rate processes using tryptophan fluorescence spectroscopy (TFS). Protein–ligand interactions are mostly associated with conformational changes in the protein structure. The TFS can follow red or blue excitation shifts of the protein tryptophan (Trp) residues, indicating coverage of the Trp by the ligand or by the conformational change. The viral proteases contain six (DENV2), seven (YFV), eight (WNV), and five (ZIKV) tryptophan residues in the complex formed between NS2B cofactor and the NS3 protease domain (NS2B/NS3^pro^). Intrinsic tryptophan (Trp) fluorescence was used to characterize the NS2B/NS3^pros^ under the influence of the EGC, EGCG, and AcEGCG molecules following the possible red or blue shift excitation changes. The maximum Trp emission on the proteins at pH 7.5 was centered at 338 nm; after titration of the EGCG and AcEGCG molecules, the emission band for DENV2, YFV, and ZIKV NS2B/NS3^pros^ shifted to 360 nm (Appendix A). This so-called red edge excitation shift (REES) is based on the Trp movement to increase polar environments [42]. On the contrary, the titration of the EGC molecule displayed no excitation shift (Appendix A), demonstrating that this molecule has a different interaction mode with the protein, which has already been demonstrated by inhibition mode experiments. Despite the high similarity between the proteases, all three molecules’ titration on the NS2B/NS3^pro^ of WNV did not induce any red or blue excitation shift of the Trp fluorescence (Appendix A).

### 3.4. In Silico Structural Analysis of Epigallocatechin Molecules with the Selected NS2B/NS3 Proteases

Extensive studies have identified potent inhibitors for flaviviruses’ NS2B/NS3^pro^. Two categories of inhibitors were identified, the catalytic site binding inhibitor and the one that binds to a different region on the protein surface; however, decreasing the protein affinity for the substrate, they are named allosteric inhibitors.

Regarding the inhibition mode experiments, the EGCG and AcEGCG are noncompetitive inhibitors, and EGC presents a competitive behavior. Since all four proteases have already been crystallized, we used the 3D model available on the Protein Data Bank (PDB) (see Section 2.8.2); those structural models seem more significant for docking and MD simulation studies. We used AutoDock Vina to dock EGCG and AcEGCG at the respective interaction site of the atomic coordinates of NS2B/NS3^pros^ (DENV2-4M9T; YFV-6URV; WNV-2IJO; ZIKV-5LC0) and EGC was docked at the active site of the proteases. The selected protease/ligand complex was subject to 100 ns (2 replicas) of MD simulation embedded in an octahedral TIP3P water box. The same starting atom position with different random initial velocities was used for the replicates. Based on binding energy and the frame cluster, a representative structure was chosen for each protease/molecule complex (Table 8).

The flexibility of the proteases in complex with the epigallocatechin through MD simulation was monitored by calculating the RMSD, RMSF, RoG, and the surface area (Appendix A). We noticed moderate deviations in the RMSD values for all NS2B/NS3^pros^ backbone atoms regarding the starting structure along the 100 ns trajectories. Concerning the complex with EGC, the NS2B domain did not stabilize well in one of the replicas along the MD simulation (Appendix A). Moreover, RMSF values for all NS2B/NS3 plus the epigallocatechin molecules atoms calculated after fitting the trajectories also showed a moderate fluctuation (Appendix A).

Besides the intermolecular H-bond time profiles (Appendix A), these results indicated that the epigallocatechin molecules (EGC and EGCG) keep forming favorable interactions with the protease residues during all of the MD simulations. On the other hand, the molecule AcEGCG maintains mostly hydrophobic interactions. Structural representation and the mode of interaction and conformation adopted by the molecules are shown in Figure 2. The EGC molecule interaction, which was docked in the active site of the proteases, was maintained mainly through hydrogen bonding interactions with key residues of the NS2B/NS3^pros^ active/binding site (Figure 2). Interestingly, the observed π-π stacking interaction for EGC complexed with NS2B/NS3^pro^ of WNV and ZIKV displaced the most stable interaction, as the π-π interaction play a pivotal role in the thermal stability.

On the other hand, the trihydroxybenzoate ring in the EGCG molecule, which is absent in the EGC molecule, stabilizes the ligand in the allosteric site through H-bonds with key amino acid residues. It is essential to observe that the absence of a trihydroxybenzoate ring completely changes the interaction mode of the molecules. Likewise, the presence of acetate groups (AcEGCG) drastically reduces the inhibition effect of the EGCG molecule.

For brevity’s sake, it is worth remembering that noncompeting molecules were suggested to bind to the pockets on the back of the protease’s active site, as described by [24,25,33].

NS2B/NS3^pro^ adopts a chymotrypsin-like fold with a catalytic triad composed of His51, Asp75, and Ser135 in the cleft between two ß-barrels. Despite the low sequence identity (~50%) (Figure 3a,b), the overall fold from DENV2, YFV, WNV, and ZIKV is very similar, with subtle divergence in the elements of the secondary structure (Figure 3c). Nevertheless, the differences adopted by the protease do not change how EGCG and AcEGCG interact in the suggested allosteric site. Figure 3d shows the allosteric binding site of the EGCG and AcEGCG molecules.

We further analyzed the electrostatic surface binding site of the complexes. Figure 4 displays the epigallocatechin molecules binding pockets by electrostatic surface of all of the studied proteases. Even if the overall RMSD of all heavy atoms is about 0.84 Å between the crystal structures of the proteases, the ligand interaction sites exhibit local geometry with a slight difference and distinct electrostatic properties. The active site (Figure 4-EGC) of all four NS2B/NS3^pros^ display polar and negatively charged surfaces; the same can be observed for the allosteric sites (Figure 4-EGCG, AcEGCG). On the contrary, the EGCG molecule binds to the ZIKV NS3B/NS3^pro^ region formed mainly by negatively charged residues (Figure 4d). For brevity’s sake, only the representative complexes regarding Table 8 were included in Figure 2 and Figure 4.

## 4. Discussion

Considering the concept of health, natural compounds that favor human and animal health would benefit, for example, in the defense against diseases transmitted by mosquitoes. Green tea catechins have a number of proven positive effects on human and animal health [43,44]. Natural molecules account for over one-third of all new, FDA-approved products, demonstrating the importance of plants and animals as a source of bioactive molecules [30]. Epigallocatechin gallate (EGCG) and epigallocatechin (EGC) are catechins and represent 59 and 19% of the total polyphenols in green tea extract [45].

This work reports the in vitro validation of promising epigallocatechin molecules against NS2B/NS3 proteases from four flaviviruses (DENV2, YFV, WNV, and ZIKV) and shows that EGC, EGCG, and its derivative EGCG octaacetate inhibit protease activity, with the EGCG showing the finest IC_50,_ in the range of 1.8 to 8 µM. It has already been described that EGCG showed antiviral activity during the early steps of infection, as well as a virucidal agent [18,20,46,47].

The mode of action of these molecules is not very well understood, but in this work, we can hypothesize that the molecules interact with the NS2B/NS3^pro^; however, in cell infection studies, they might interact with multiple binding sites [48]. In this sense, in vitro studies were performed, and the inhibition mode demonstrated the probable interaction site of the molecules on the NS2B/NS3^pro^. Molecular docking and dynamics simulation studies predicted the epigallocatechin molecule’s binding pocket, showing that the EGC potentially forms interactions with the activity site of the studied proteins. On the other hand, EGCG and AcEGCG molecules interact with the viral protease allosterically.

Despite the polymorphism identified in several viral proteins, this modification may not affect the catalytic efficiency of the enzyme [49,50]. Therefore, further studies are needed to confirm this; however, it seems reasonable to assume the inhibition and interaction properties of the molecules observed in this study.

The molecule EGC exhibited a competitive inhibition model suggesting that even if the molecule interacts with amino acid residues in the active/binding site, it still allows the interaction between the substrate and the protein, probably regarding the size of the molecule.

In our in vitro assay, the EGCG molecule interacts noncompetitively with the proteases, indicating that this molecule is an allosteric inhibitor. We suggest the trihydroxybenzoate ring prevents the molecule from interacting with the active site. Yadav and colleagues [51] described the interaction of ZIKV NS2B/NS3^pro^ with the EGCG molecule through the active site using in silico studies, which disagrees with our experimental results, which show that this molecule is an allosteric inhibitor. In addition to their virucidal capacity, several studies have shown that EGCG can act by blocking the necessary receptors for viral attachment to the target cell, as in the case of HIV, where the drug binds to CD4 expressed on T cells and inhibits the virus binding to this site, blocking viral infection [20,52]. Carneiro and coll. [20] described no influence of the molecule on viral infection, using a pre-treatment assay, which suggested that EGCG has no action on the expression or in blocking the cellular receptors used by ZIKV during the entry process in the host cell. Vázquez-Calvo and colleagues [18] described that the EGCG molecule directly affects WNV particles exerting a virucidal effect and reducing the effectivity of ZIKV and DENV. The inhibitory capacity of green tea catechins has also been described for SARS-CoV-2 main protease [18,49], HasV endopeptidase [50], HCV NS3/4A protease [51], and HIV-1 protease [52].

Here, we have demonstrated that the epigallocatechin molecules presented in this study, natural compounds found in abundance in many foods (EGC and EGCG) and its modified form (AcEGCG), inhibit in vitro the activity of the essential virus protease (NS2B/NS3). Despite these encouraging results, we have shown that combining the EGC and EGCG molecules, which displayed different interaction modes (competitive and noncompetitive), drastically increased the inhibition of the YFV, WNV, and ZIKV proteases, displaying an IC_50_ value in the nM range.

EGCG and its derivatives have a range of wide biological activity, but in the human body, these molecules can undergo multiple processes of metabolization [53,54,55]. Temperature, pH, and oxidant concentration accelerate the degradation rate [56]. Green tea catechins are very unstable in the saliva, stomach, and upper intestine [57]. The intestinal microflora has the potential to dissociate and degrade EGCG in vitro and in vivo [58]. The chemical instability of green tea catechins reduces intestinal absorption and affects bioavailability. However, several studies have demonstrated the improvement of bioaccessibility by nanoliposome encapsulation [57], structural modification [55], methylation [59], cyclisation [60], or glycosidation [61]. Even though epigallocatechin molecules are derived from food plants, high doses have side effects on the human body; for example, green tea catechins can trigger anxiolytic reactions, hypoglycemic activity, hypochromic anemia, and liver and kidney failure. However, these effects are not related to the host serine proteases [62,63].

In conclusion, our results have revealed the antiviral activity of two different epigallocatechin, EGC and EGCG, and its derivative, AcEGCG, against the protease of four clinically relevant members of the *Flavivirus* genus (DENV2, YFV, WNV, and ZIKV) and point to EGC and EGCG as potential scaffolds for the prospect design of mixed antiviral compounds that affect the viral protein in two different binding regions. Optimization of these bioactive molecules is essential to achieve high oral bioavailability, including chemical modifications, absorption, and penetration enhancers, as well as formulation design. It is important to emphasize that oral delivery of new lead molecules can be facilitated by a thorough understanding of the interaction of the inhibitory molecule with its target.

## Figures and Tables

**Figure 1 pharmaceutics-15-00803-f001:**
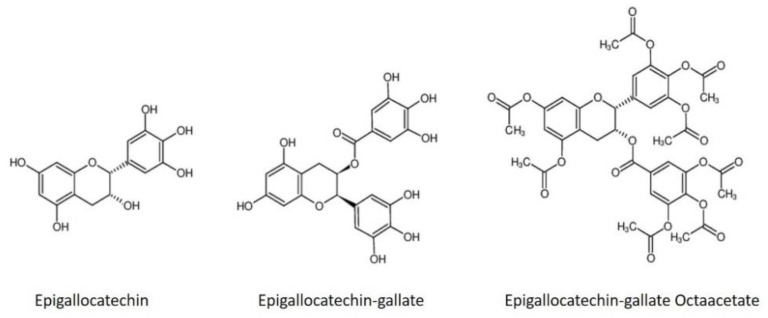
Structure of green tea catechins and their synthetic derivatives. Epigallocatechin, epigallocatechin gallate, and epigallocatechin gallate octaacetate.

**Figure 2 pharmaceutics-15-00803-f002:**
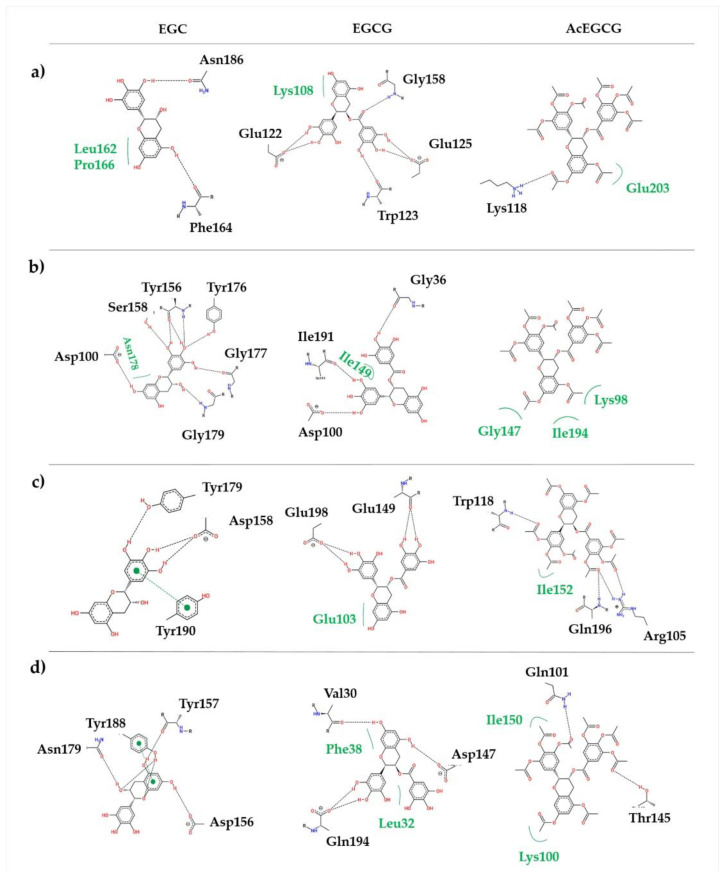
Epigallocatechin molecules binding mode and conformation of the representative complex. (**a**) DENV, (**b**) YFV, (**c**) WNV, and (**d**) ZIKV.

**Figure 3 pharmaceutics-15-00803-f003:**
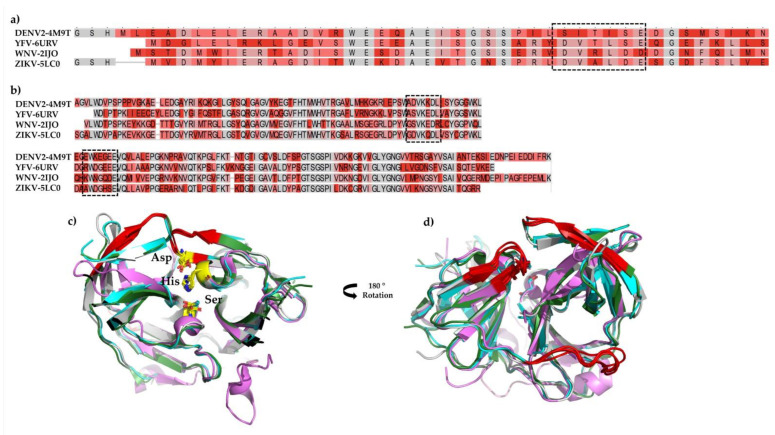
Sequence alignment of the NS2B (**a**) and NS3pro (**b**) of the studied viruses. Structural alignment highlighting the catalytic triad (**c**) and the allosteric binding site (**d**).

**Figure 4 pharmaceutics-15-00803-f004:**
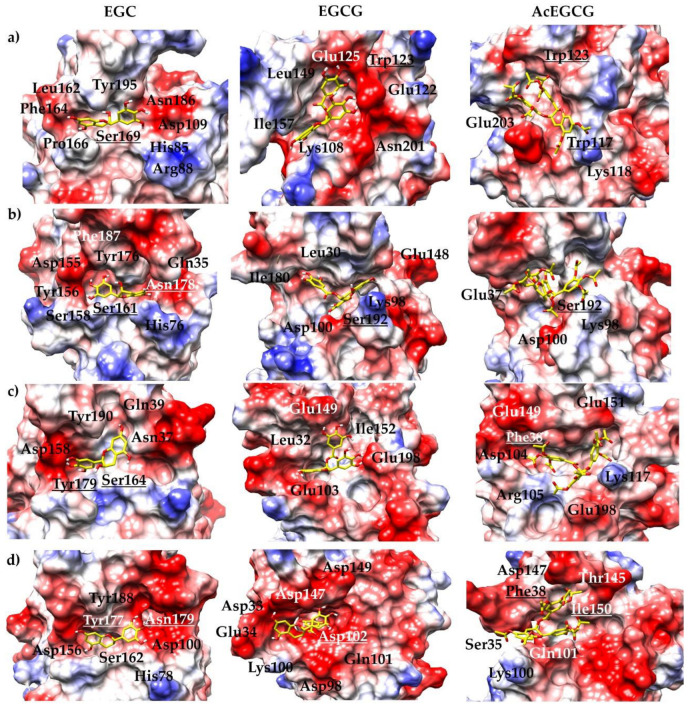
Electrostatic surface potentials of the representative model. The figure is colored red and blue for negative and positive charges, respectively, and white represents neutral residues. The epigallocatechin molecules are shown as yellow sticks. The proteases of (**a**) DENV, (**b**) YFV, (**c**) WNV, and (**d**) ZIKV are shown by the electrostatic surface.

**Table 1 pharmaceutics-15-00803-t001:** Protein, substrate, and epigallocatechin molecules concentration used in the activity assay.

Virus	Protein (nM)	Substrate (µM)	EGC (µM)	EGCG (µM)	AcEGCG (µM)
DENV2	250	50	0–900	0–200	0–200
YFV	6	50	0–200	0–200	0–200
WNV	50	50	0–250	0–125	0–500
ZIKV	3	20	0–120	0–50	0–500

**Table 2 pharmaceutics-15-00803-t002:** Protein, substrate, and epigallocatechin molecules concentration used in the inhibition mode assay.

Virus	Protein (nM)	Substrate (µM)	EGC (µM)	EGCG (µM)	AcEGCG (µM)
DENV2	250	35–100	0, 5, 10, 15	0, 0.5, 1, 2.5	0, 10, 20, 30
YFV	6	30–90	0, 1, 5, 10	0, 1, 2, 4	0, 10, 20, 40
WNV	50	20–80	0, 5, 10, 25	0, 0.5, 1, 1.5	0, 10, 25, 50
ZIKV	3	5–40	0, 1, 5, 10	0, 0.5, 1, 2	0, 5, 10, 20

**Table 3 pharmaceutics-15-00803-t003:** Epigallocatechin molecules concentration range used in fluorescence spectroscopy.

	Concentration (µM)
**DENV2**	
EGCG	0–60
EGC	0–62
AcEGCG	0–200
**YFV**	
EGCG	0–78
EGC	0–94
AcEGCG	0–260
**ZIKV**	
EGCG	0–56
EGC	0–70
AcEGCG	0–208
**WNV**	
EGCG	0–22
EGC	0–84
AcEGCG	0–200

**Table 4 pharmaceutics-15-00803-t004:** Assay condition for the inhibition test.

Virus	Protein (nM)	Substrate (µM)	EGCG (µM)	EGC (µM)	AcEGCG (µM)
DENV2	250	50	0–200	0–900	0–200
YFV	6	50	0–200	0–200	0–200
ZIKV	3	20	0–50	0–120	0–500
WNV	50	50	0–125	0–250	0–500

**Table 5 pharmaceutics-15-00803-t005:** IC_50_ values and inhibition mode of the epigallocatechin molecules.

Virus Protease	Molecule	IC_50_ [µM]	Inhibition Mode
	EGCG	6.3 ± 1.0	Noncompetitive
DENV2	EGC	145 ± 3.0	Competitive
	AcEGCG	39.8 ± 2.5	Noncompetitive
	EGCG	8 ± 1.4	Noncompetitive
YFV	EGC	19 ± 1.9	Competitive
	AcEGCG	39.1 ± 2.2	Noncompetitive
	EGCG	4.5 ± 0.6	Noncompetitive
ZIKV	EGC	8.9 ± 1.2	Competitive
	AcEGCG	102 ± 2.6	Noncompetitive
	EGCG	1.8 ± 0.5	Noncompetitive
WNV	EGC	17.3 ± 2.5	Competitive
	AcEGCG	83.4 ± 10.4	Noncompetitive

**Table 6 pharmaceutics-15-00803-t006:** IC_50_ values of the combined EGC-EGCG molecules.

NS2B/NS3	IC_50_ [µM]
DENV2	35.1 ± 3.2
YFV	1.17 ± 0.2
ZIKV	0.57 ± 0.05
WNV	0.58 ± 0.07

**Table 7 pharmaceutics-15-00803-t007:** K_D_ values of the molecules in complex with the NS2B/NS3^pros^ by SPR experiments.

	K_D_ (µM) ± STD		
	EGCG	EGC	AcEGCG
DENV2	0.07	0.03	2.4
YFV	0.06	0.27	0.47
ZIKV	0.09	0.02	4.8
WNV	0.07	0.22	0.98

**Table 8 pharmaceutics-15-00803-t008:** Summary of the MD simulations of the NS2B/NS3 proteases in complex with the epigallocatechin molecules.

NS2B/NS3^pro^	Molecules	Binding Energy (kcal/mol)		Representative Frame Cluster (%)	
		Replicate 1	Replicate 2	Replicate 1	Replicate 2
DENV2	EGCG	−30.7 ± 4.7	−31.9 ± 3.9	76.3	51.7
	EGC	−28.4 ± 5.7	−25.5 ± 3.1	53	94
	AcEGCG	−48.5 ± 4.3	−38.7 ± 3.8	78.6	83.4
YFV	EGCG	−28.1 ± 4.8	−25.3 ± 4.2	91.2	87.1
	EGC	−23.6 ± 5.8	−24.6 ± 5.6	65.2	81.8
	AcEGCG	−25.7 ± 4.5	−40.4 ± 3.7	87.7	85.9
WNV	EGCG	−38.1 ± 5.4	−25.1 ± 5.2	68.8	96.1
	EGC	−21.4 ± 3.2	−25.8 ± 4.8	89	78.2
	AcEGCG	−38.8 ± 4.4	−43.5 ± 3.5	83.6	77.1
ZIKV	EGCG	−22.7 ± 5.8	−36.0 ± 6.0	59.8	26.3
	EGC	−26.0 ± 2.9	−27.0 ± 3.5	57.4	74.7
	AcEGCG	−27.9 ± 5.0	−31.2 ± 4.5	91.3	41.1

## Data Availability

Data are contained within the article and Appendix A.

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
