# Peer review of "The Importance of Epigallocatechin as a Scaffold for Drug Development against Flaviviruses"

_pharmaceutics, 2023, doi:10.3390/pharmaceutics15030803_

Round 1

Reviewer 1 Report

This manuscript describes the author's work in investigation of epigallocatechins against flaviviruses such as DENV, WNV, and ZIKV. While the epigallotins do have activity against the Flaviviridae their activity is rather low being > 50 microM and these phenolic derivatives have horrible pharmacokinetics being rapidly glucuronidated. Still the study is interesting and describes the potency of the title molecules against the viral NS3 gene product and is supported by some computational work. Interestingly the authors investigated epigallotin octoacetate. In vivo this molecule will not be metabolically stable and would surely be subjected to rapid cleavage of the acyl moieties by endogenous esterases. This paper could be published in that perhaps the results could be used as the basis for improvement of the epigallotins activities against the target viruses and the enzyme activity studies are interesting.

Author Response

The response file is attached.

Reviewer 2 Report

The manuscript submitted by Coronado et al reports the antiviral activity of two polyphenols molecules i.e., EGC and EGCG against NS2B/NS3 proteases found in flaviviruses such as Dengue, Yellow fever virus, West nile virus and Zika. This manuscript is well structured overall, but it needs minor modifications, that I have outlined in the comments below.

1.     Line 35: Please add IC50 values.

2.     Line 47-51: Please add reference to the claim made in this sentence.

3.     Line 107-109: Please fix the grammar of this sentence.

4.     Table 1: The authors have to explain why different concentration of protein, substrate, and epigallocatechin was used in the enzymatic assay.

5.     Line 128-129: Please mention different concentration of molecules in the manuscript.

6.     Discussion: This section is very weak; the authors have to discuss more about their finding and rational behind each experiment they have performed in this manuscript.

7.     Figure 3: It is very hard to visualize, please add high resolution figure.

8.     The authors have to talk about the limitations of these studies. How this can be translated into human studies? What challenges can be expected?

9.     Why was control not used for in vitro biological assays?

10.  Conclusion: This section is too short, please elaborate about the finding.

11.  The authors have to proofread the manuscript, there are grammatical errors.

Author Response

The response file is attached

Reviewer 3 Report

The present article analyses the mechanism of action of EGC, EGCG and AcEGCG in targeting NS2B/NS3 proteinases of four members of Flaviviridae family, namely YFV, DENV-2, WNV and ZIKV.

Thus, the compounds have provided a broad spectrum of inhibitory activities against these enzymes, with potencies in the low micromolar range for EGC, EGCG. The investigation of the enzyme inhibition disclosed a different mode, competitive for EGC and non-competitive for the other two molecules. The combination of EGC and EGCG led to an improved inhibitory profile versus the target enzymes. However, none of the compounds was tested in cell-based assays against the viruses here considered.

The paper is well organised and written. However, it needs to be improved as follows:

- Since these viral infections are vector-borne diseases, in the introduction the One Health concept that recognizes the interdependent relationship between human health, animal health and environment, has to be discussed. The drug R&D expertise for both human and animal parasitic diseases has to be integrated.

- In the introduction please cite and describe the review where a good example of competitive and non-competitive inhibitors of these enzymes are reported, as a background to this study.

Samrat SK, Xu J, Li Z, Zhou J, Li H. Antiviral Agents against Flavivirus Protease: Prospect and Future Direction. Pathogens. 2022 Feb 25;11(3):293. doi: 10.3390/pathogens11030293.

- Please also discuss in the paper the molecular analysis and polymorphism of NS2B-NS3 protease from YFV and other flaviviruses!

See Noske GD, Gawriljuk VO, Fernandes RS, Furtado ND, Bonaldo MC, Oliva G, Godoy AS. Structural characterization and polymorphism analysis of the NS2B-NS3 protease from the 2017 Brazilian circulating strain of Yellow Fever virus. Biochim Biophys Acta Gen Subj. 2020 Apr;1864(4):129521. doi: 10.1016/j.bbagen.2020.129521

- provide a comparative analysis of the similarities and differences of NS2B-NS3 protease from the four flaviviruses. This could help understand how the drug design could move towards the development of new molecules, not only based on ECG scaffold.

- It should also be relevant to deal with the concept of protease mutations in the context of the drug resistance issue.

-  Regarding the inhibitory activity of ECG against the four proteases that make candidate EGC as a potential broad spectrum anti flavivirus agent it is also important taking into consideration that viral NS2B-NS3 proteases are similar to other host serine proteases. Therefore, the risk of adverse effects could be high. Please look into this issue.

Author Response

The response file is attached.

Round 2

Reviewer 3 Report

The paper has been properly improved according to the suggestions proposed. Now I consider this paper of interest for researchers working in the field and I suggest its publication in its current form